# Conditional quantum operation of two exchange-coupled single-donor spin qubits in a MOS-compatible silicon device

Mateusz T. Mądzik [1], Arne Laucht [1], Fay E. Hudson [1], Alexander M. Jakob[2], Brett C. Johnson[2], David N. Jamieson [2], Kohei M. Itoh [3], Andrew S. Dzurak [1] & Andrea Morello [1✉]

Silicon nanoelectronic devices can host single-qubit quantum logic operations with fidelity better than 99.9%. For the spins of an electron bound to a single-donor atom, introduced in the silicon by ion implantation, the quantum information can be stored for nearly 1 second. However, manufacturing a scalable quantum processor with this method is considered challenging, because of the exponential sensitivity of the exchange interaction that mediates the coupling between the qubits. Here we demonstrate the conditional, coherent control of an electron spin qubit in an exchange-coupled pair of $^{31}$P donors implanted in silicon. The coupling strength, $J = 32.06 \pm 0.06$ MHz, is measured spectroscopically with high precision. Since the coupling is weaker than the electron-nuclear hyperfine coupling $A \approx 90$ MHz which detunes the two electrons, a native two-qubit controlled-rotation gate can be obtained via a simple electron spin resonance pulse. This scheme is insensitive to the precise value of $J$, which makes it suitable for the scale-up of donor-based quantum computers in silicon that exploit the metal-oxide-semiconductor fabrication protocols commonly used in the classical electronics industry.

[1] Centre for Quantum Computation and Communication Technology, School of Electrical Engineering and Telecommunications, UNSW Sydney, Sydney, NSW 2052, Australia. [2] Centre for Quantum Computation and Communication Technology, School of Physics, University of Melbourne, Melbourne, VIC 3010, Australia. [3] School of Fundamental Science and Technology, Keio University, 3-14-1, Hiyoshi 223-8522, Japan. ✉email: a.morello@unsw.edu.au

**B**uilding useful quantum computers is a challenge on many fronts, from the development of quantum algorithms[1] to the manufacturing of scalable hardware devices[2]. For the latter, adapting the fabrication processes already in use in the classical electronics industry—silicon-based metal-oxide-semiconductor (MOS) processing[3–6] and ion implantation[7–9]—to the construction of quantum hardware would represent a great technological head start. This was the insight that triggered the first proposal of encoding quantum information in the spin state of donor atoms in silicon[10]. Qubits defined by individual donor-bound electron spins have demonstrated quantum gate fidelities beyond 99.9% (ref. [11]), and coherence lifetimes approaching 1 s (ref. [12]). The next challenge is the demonstration of robust two-qubit logic operations, necessary for universal quantum computing. In this work, we demonstrate the key capability of performing conditional, coherent quantum operations on single-donor spin qubits in the presence of weak exchange interaction[13]. The weak interaction regime is crucial to ensure a mode of operation that is compatible with the inherent manufacturing tolerances of silicon MOS devices.

In their simplest form, two-qubit logic gates can be executed using three distinct strategies. The first requires the two qubits to have approximately the same energy splitting, $\epsilon_1 \approx \epsilon_2$, and turning on the qubit–qubit interaction $J$ for a finite amount of time[14], yielding a native SWAP gate[15]. The second strategy implements a controlled-Z gate by dynamical control of $J$. The coupling is switched on for a calibrated time period, whereby the target qubit acquires a phase shift proportional to the change in precession frequency determined by the state of the control qubit[16,17]. The third strategy implements a native controlled-rotation (CROT) gate via resonant excitation of the target qubit, whose transition frequency can be made to depend on the state of the control qubit.

The CROT gate is related to the controlled-NOT operation that appears in most quantum algorithms, but imparts an additional phase of $\pi/2$ to the target qubit. This gate requires the individual qubits' energy splittings to differ by an amount $\delta\epsilon = |\epsilon_1 - \epsilon_2|$ much larger than their coupling $J$. It was used in early nuclear magnetic resonance (NMR) experiments[18], superconducting qubits[19] and, more recently, was adapted to electron spin qubits in semiconductors, where the energy detuning $\delta\epsilon$ can be provided by a difference in Landé g-factors between the two electron spins[16,20] or by a magnetic field gradient[21]. For electron spin qubits, the coupling $J$ originates from the Heisenberg exchange interaction. The main advantage of this type of gate is that it can be performed while keeping $J$ constant—an essential feature when locally tuning $J$ is either impossible or impractical. Moreover, the precise value of $J$ is unimportant, as long as it is smaller than $\delta\epsilon$, and larger than the resonance linewidth.

For donor electron spin qubits in silicon, two-qubit logic gates based on exchange interactions are particularly challenging. Because of the small ($\approx 2$ nm) Bohr radius of the electron wave function[22], the exchange interaction strength decays exponentially with distance and, when accounting for valley interference, it can even oscillate upon displacing the atom by a single lattice site[23]. Therefore, a two-qubit CROT gate where $J$ can be kept constant and does not need to have a specific value (within a certain range), is highly desirable. An embodiment of such gate was proposed by Kalra et al.[13], who recognized that the energy detuning $\delta\epsilon$ between two donor electrons can be provided in a convenient and natural way by setting the donor nuclear spins in opposite states. This causes the electron spins' energy splittings to differ by the electron-nuclear hyperfine coupling $A \approx 100$ MHz.

Until now, all experimental observations of exchange coupling between individual pairs of donors have been obtained in the regime $J \gg 100$ MHz (refs. [24–27]), where the native CROT gate described above cannot be performed. A SWAP operation was recently demonstrated between strongly exchange-coupled electron spins bound to donor clusters[27], albeit without coherent quantum control of the individual spins. Here, we present the experimental observation of weak exchange interaction in a pair of $^{31}$P donors, and the coherent operation of one qubit conditional on the state of the other. Achieving these results with ion-implanted donors in a MOS device (Fig. 1) reaffirms the applicability of standard semiconductor manufacturing methods to silicon-based quantum computing.

## Results

**Engineering conditional quantum logic operations with weak exchange.** The operating principle of a two-qubit CROT operation for $^{31}$P donors in the presence of weak exchange coupling $J$ can be understood from their spin Hamiltonian:

$$H = (\mu_B/h)B_0(g_t S_{zt} + g_c S_{zc}) + \gamma_n B_0 (I_{zt} + I_{zc}) \\ + A_t \mathbf{S}_t \cdot \mathbf{I}_t + A_c \mathbf{S}_c \cdot \mathbf{I}_c + J(\mathbf{S}_t \cdot \mathbf{S}_c), \quad (1)$$

The donors are placed in a static magnetic field $B_0$ ($\approx 1.4$ T in our experiment) and their spins are described by the electron ($\mathbf{S}_t$, $\mathbf{S}_c$, with basis states $|\uparrow\rangle, |\downarrow\rangle$) and nuclear ($\mathbf{I}_t$, $\mathbf{I}_c$, with basis states $|\Uparrow\rangle, |\Downarrow\rangle$) spin 1/2 vector Pauli operators; the subscripts "c" and "t" refer to the control and target qubit, respectively. $\mu_B$ is the Bohr magneton, $h$ is the Planck constant, and $g_t$, $g_c \approx 1.9985$ are the Landé g-factors, such that $g\mu_B/h \approx 27.97$ GHz/T. The nuclear gyromagnetic ratio is $\gamma_n \approx -17.23$ MHz/T, and $A_t$, $A_c$ are the electron-nuclear contact hyperfine interactions in the target and

**a**

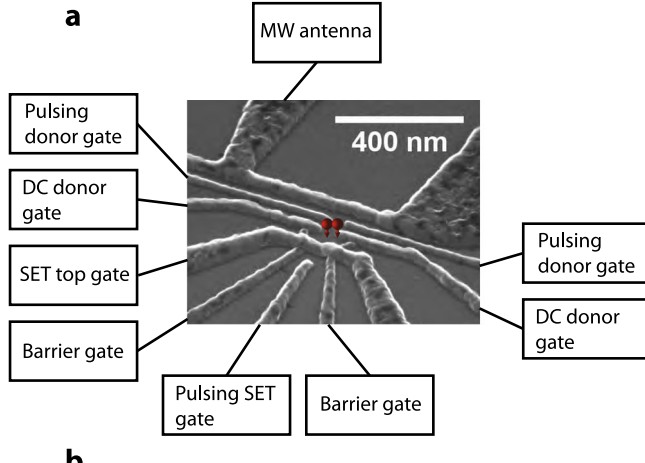

- MW antenna
- Pulsing donor gate
- DC donor gate
- SET top gate
- Barrier gate
- Pulsing SET gate
- Barrier gate
- Pulsing donor gate
- DC donor gate

**400 nm**

**b**

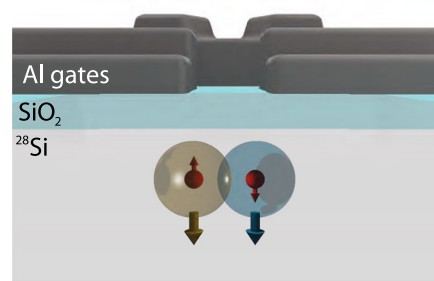

Al gates
SiO₂
$^{28}$Si

**Fig. 1 Two-qubit metal-oxide-semiconductor device. a** Scanning electron micrograph of a device similar to the one used in the experiment, with labels describing the function of the aluminum gates on the surface. **b** Schematic cross section of the device, depicting a pair of donors $\approx 10$ nm beneath a thin SiO₂ dielectric, inside an isotopically enriched $^{28}$Si epilayer.

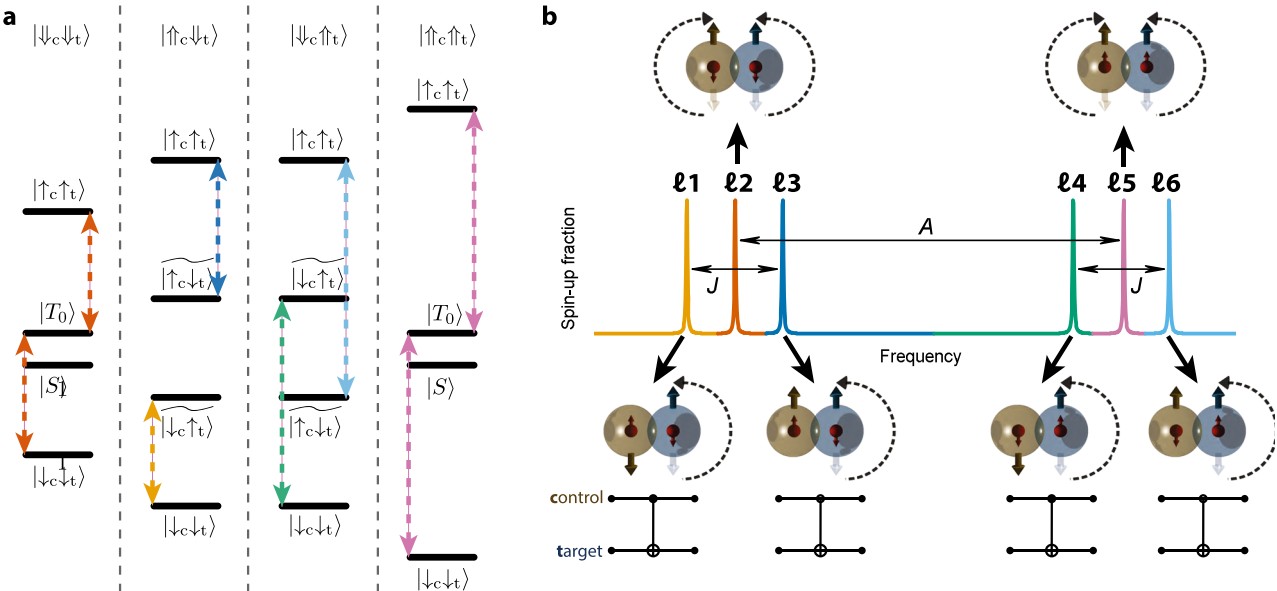

**Fig. 2 Two-qubit gate operation for weakly exchange-coupled $^{31}$P electron spin qubits. a** Electronic energy level diagram of a pair of $^{31}$P donors in the four possible nuclear spin configurations; we assume here for simplicity $A_c = A_t = A$ and $J \ll A$. **b** Simplified schematic of the ESR spectrum of the target electron in the four possible nuclear spin configurations, and two control electron spin orientations. The nuclear spins provide an energy detuning $\delta\epsilon \equiv A$, while the exchange interaction splits by $J$ the resonance frequencies of the target qubit, depending on the state of the control qubit. At the bottom, cartoons and quantum circuit diagrams illustrate the electron spin rotations and the quantum gate operations (CROT and zero-CROT) obtainable on each of the depicted resonance lines.

in the control donor, respectively; their average is $\bar{A} = (A_t + A_c)/2$ and their difference $\Delta A = (A_t - A_c)$.

To simplify the problem, we draw the energy levels diagrams shown in Fig. 2a, where we assume that both donors have the same hyperfine coupling $A \approx 100$ MHz. A more general and extensive discussion of the two-donor spin Hamiltonian is given in the Supplementary Note 1.

When the nuclei are in a parallel configuration ($|\Downarrow_c \Downarrow_t\rangle$ or $|\Uparrow_c \Uparrow_t\rangle$), the uncoupled electron spins have the same energy splitting. Upon introducing an exchange coupling $J$, the electronic eigenstates become the singlet $|S\rangle = (|\downarrow_c \uparrow_t\rangle - |\uparrow_c \downarrow_t\rangle)/\sqrt{2}$ and triplet $|T_-\rangle = |\downarrow_c \downarrow_t\rangle, |T_0\rangle = (|\downarrow_c \uparrow_t\rangle + |\uparrow_c \downarrow_t\rangle)/\sqrt{2}, |T_+\rangle = |\uparrow_c \uparrow_t\rangle$ states. An oscillating magnetic field can induce electron spin resonance (ESR) transitions between the triplets, corresponding to the ESR lines $\ell2$ and $\ell5$ in Fig. 2b. The singlet state has a total spin of zero, and cannot be accessed by ESR. Since the energy splittings $|T_-\rangle \leftrightarrow |T_0\rangle$ and $|T_0\rangle \leftrightarrow |T_+\rangle$ are identical, an ESR transition can occur irrespective of the state of the control qubit. These unconditional resonances do not constitute two-qubit logic operations.

If instead, we prepare the nuclear spins in opposite orientations ($|\Downarrow_c \Uparrow_t\rangle$ or $|\Uparrow_c \Downarrow_t\rangle$), the hyperfine interaction detunes the uncoupled electrons by $\delta\epsilon \equiv A$. Introducing a weak exchange coupling $J \ll A$ results in electronic eigenstates of the form $|\downarrow_c \downarrow_t\rangle, |\widetilde{\uparrow_c \downarrow_t}\rangle, |\widetilde{\downarrow_c \uparrow_t}\rangle, |\uparrow_c \uparrow_t\rangle$, where $|\widetilde{\uparrow_c \downarrow_t}\rangle = \cos\theta|\uparrow_c \downarrow_t\rangle + \sin\theta|\downarrow_c \uparrow_t\rangle$, $|\widetilde{\downarrow_c \uparrow_t}\rangle = \cos\theta|\downarrow_c \uparrow_t\rangle - \sin\theta|\uparrow_c \downarrow_t\rangle$, and $\tan(2\theta) = J/A$. In this case, for each antiparallel nuclear orientation there exist two distinct frequencies ($\ell1$ and $\ell3$ for $|\Uparrow_c \Downarrow_t\rangle$, $\ell4$ and $\ell6$ for $|\Downarrow_c \Uparrow_t\rangle$), separated by $J$, at which the target qubit would respond, depending on the state of the control. Therefore, a $\pi$-pulse on any of these resonance lines embodies a form of two-qubit CROT gate. Defining $|\downarrow\rangle$ as the computational $|1\rangle$ state, $\ell1$ and $\ell4$ yield CROT gates, i.e., rotations of the target qubit conditional on the control being in the $|1\rangle$ state, while $\ell3$ and $\ell6$ yield zero-CROT gates (Fig. 2b).

Importantly, the ability to perform a CROT gate depends only on the ability to apply a selective $\pi$-pulse on one of the conditional resonances. The precise value of $J$ is unimportant, as long as it exceeds the resonance linewidth (~10 kHz in our devices) and is smaller than $A \approx 100$ MHz. The value of $J$ sets an approximate limit to the speed of the CROT operation[13], since the spectral width of the CROT pulse (approximately equal to the Rabi frequency) must be lower than the frequency spacing $= J$ between resonances conditioned on opposite control qubit states[20]. This limitation can be circumvented to some extent by replacing the simple resonant $\pi$-pulse with more sophisticated control schemes[28,29], as demonstrated, e.g., in quantum dot systems[20]. Overall, this scheme affords a wide tolerance in the physical placement of the donors.

**Ion implantation strategies**. We fabricated two batches of devices designed to exhibit exchange interaction between donor pairs. In addition to the implanted $^{31}$P donors, the devices include a single-electron transistor (SET) to detect the donor charge state, four electrostatic gates to control the donor potential, and a microwave antenna to deliver oscillating magnetic fields (see Fig. 1a).

The ion implantation step was executed using two different strategies. We first implanted a batch of devices with a low fluence of $P_2^+$ molecular ions, accelerated with a 20 keV voltage (corresponding to 10 keV/atom). When a $P_2^+$ molecule hits the surface of the chip, the two P atoms break apart and come to rest at an average distance that depends on the implantation energy. We chose the energy and the fluence ($5 \times 10^{10}$ donors/cm$^2$) to obtain well-isolated pairs; that is, we used the choice of acceleration energy to determine the most likely distance between donors resulting from an individual $P_2^+$ molecule (see Fig. 3c), and adapted the fluence to obtain a low probability of donor pairs overlapping with each other. A representative charge stability diagram of this type of devices, taken by sweeping the SET top gate voltage, stepping the donor gate voltage, and monitoring the

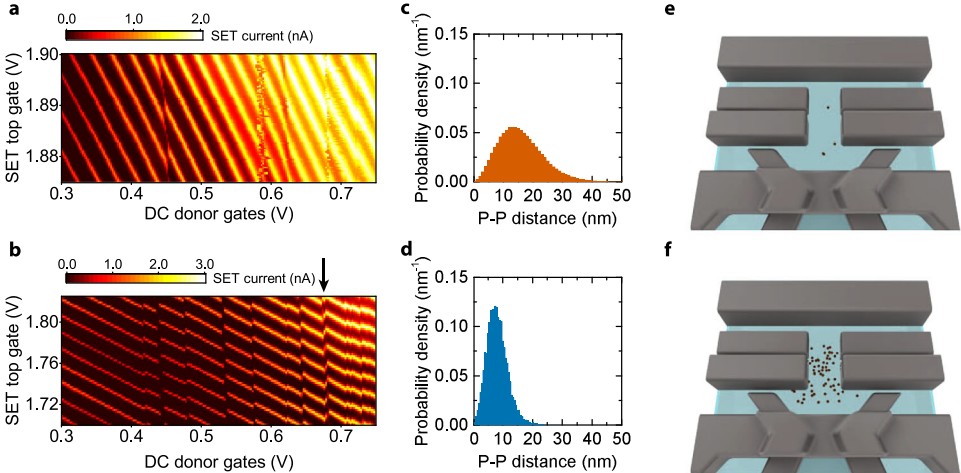

**Fig. 3 Comparison of two ion implantation strategies. a**, **b** The current through a single-electron transistor (SET) displays characteristic Coulomb peaks, appearing as bright diagonal lines, as a function of the gate voltages. The presence of a donor coupled to the SET is revealed by discontinuities in the pattern of Coulomb peaks, occurring when the donor changes its charge state. **a** Charge stability diagram (i.e., SET current vs. SET and donor gates voltages) in a device where $P_2^+$ molecular ions were implanted at a fluence corresponding to $5 \times 10^{10}$ donors/cm$^2$, compared to **b** a device where $P^+$ single ions were implanted with high fluence, yielding $1.25 \times 10^{12}$ donors/cm$^2$. The much higher number of observable charge transitions in **b** is consistent with the higher donor density in the device. An arrow indicates a region where the charge transitions of two different donors cross each other (see also Fig. 4a). **c** Simulated probability density of inter-donor distance for $P_2^+$ molecule implantation at the fluence of $5 \times 10^{10}$ donors/cm$^2$. **d** A much higher probability density for small inter-donor distances is obtained for $P^+$ implantation at the fluence $1.25 \times 10^{12}$ donors/cm$^2$. The device sketches show simulated random placements of donors for the $P_2^+$ molecular (**e**), and the high-fluence $P^+$ ion (**f**) implantation strategies. Red dots represent $P^+$ ions that crossed through the 8 nm thick SiO$_2$ dielectric layer and stopped in the Si crystal, thus becoming active substitutional donors.

transistor current, is shown in Fig. 3a. A small number of isolated donor charge transitions—identifiable as near-vertical breaks in the regular patterns of SET current peaks—reveals well-separated individual donors, but too low a chance that two donors may be found in close proximity.

We thus fabricated another batch of devices, where we implanted a high fluence ($1.25 \times 10^{12}$ donors/cm$^2$) of single $P^+$ ions at 10 keV energy. This yields a 25-fold increase in the donor density (see Fig. 3d, f and Supplementary Note 2), reflected in the much larger number of observed charge transitions in a typical stability diagram (Fig. 3b).

In a device with high-fluence $P^+$ implanted donors, we identified a pair of charge transitions that, under suitable gate tuning, cross each other (Fig. 4a). As expected from the electrostatics of double quantum dots, this results in a "honeycomb diagram", where the crossing between the charge transitions is laterally displaced by the mutual charging energy of the two donors[30]. Note that this in itself does not provide any indication of the existence of a quantum-mechanical exchange coupling. Spin exchange would appear as a curvature in the sides of the honeycomb diagram[31], but its value would need to be $\gg 1$ GHz to be discernible in this type of experiment.

**Spectroscopic measurement of exchange interaction**. The experimental methods for control and readout of the $^{31}$P donors follow well-established protocols. We perform single-shot electron spin readout via spin-dependent tunneling into a cold charge reservoir[32,33], and coherent control of the electron[34] and nuclear[35] spins via magnetic resonance, where an oscillating magnetic field is provided by an on-chip broadband microwave antenna[36].

Controlling the two pulsing gates above the donor implantation area allows us to selectively and independently control the charge state of each donor, which can be set to either the neutral $D^0$ (electron number $N = 1$) or the ionized $D^+$ ($N = 0$) state. In particular, we can freely choose the electrochemical potential of the donors with respect to each other, i.e., which of the donors

ionizes first, while the other remains neutral (see Supplementary Movie).

On the stability diagram in Fig. 4a, we identify the four regions corresponding to the neutral ($N = 1$) and ionized ($N = 0$) charge states of each donor. For example, the boundary between the $(0_c, 0_t)$ and $(0_c, 1_t)$ regions is where the second donor (target) can be read out via spin-dependent tunneling to the SET island[32,33], while the first (control) remains ionized. This is because, when transitioning from, e.g., $(0_c, 1_t)$ to $(0_c, 0_t)$, the lost charge is absorbed by the island of the SET, which is tunnel coupled to the donors[32]. At low electron temperatures ($T_{el} \approx 100$ mK) and in the presence of a large magnetic field $B_0 \approx 1.4$ T, the tunneling of charge from donor to SET island becomes spin dependent, since only the $|\uparrow\rangle$ state has sufficient energy to escape from the donor. This mechanism provides the basis for the single-shot qubit readout[33]. Therefore, the boundary $(0_c, 0_t) \leftrightarrow (0_c, 1_t)$ is where we can observe the spin target donor, while it behaves as an isolated system, since the control donor is ionized at all times.

This expectation is confirmed by the ESR spectrum shown in Fig. 4b, which exhibits the two ESR peaks consistent with the two possible nuclear spin orientations of a single $^{31}$P donor[35]. Since we are measuring a single atom, each trace normally contains only one peak, but occasionally the nuclear spin flips direction during the scan, so a single trace can also exhibit both peaks. Since the intrinsic ESR linewidth is very narrow (a few kilohertz in isotopically enriched $^{28}$Si (ref. [12])), finding the resonances is a time-consuming process. To speed this up, we used adiabatic spin inversion[37] with a 6 MHz frequency chirp, resulting in a large electron spin-up fraction whenever a resonance falls within the frequency sweep range. The 6 MHz width of the frequency sweeps is the cause of the artificial width and shape of the resonances shown in Fig. 4b, c.

In the next step, we operate near the boundary $(1_c, 0_t) \leftrightarrow (1_c, 1_t)$ where the target donor is read out, but the control donor is in the neutral $D^0$ charge state, with an electron bound to it. Repeatedly measuring the ESR spectrum of the target donor, now reveals four possible ESR peaks. We interpret this as evidence for the presence of

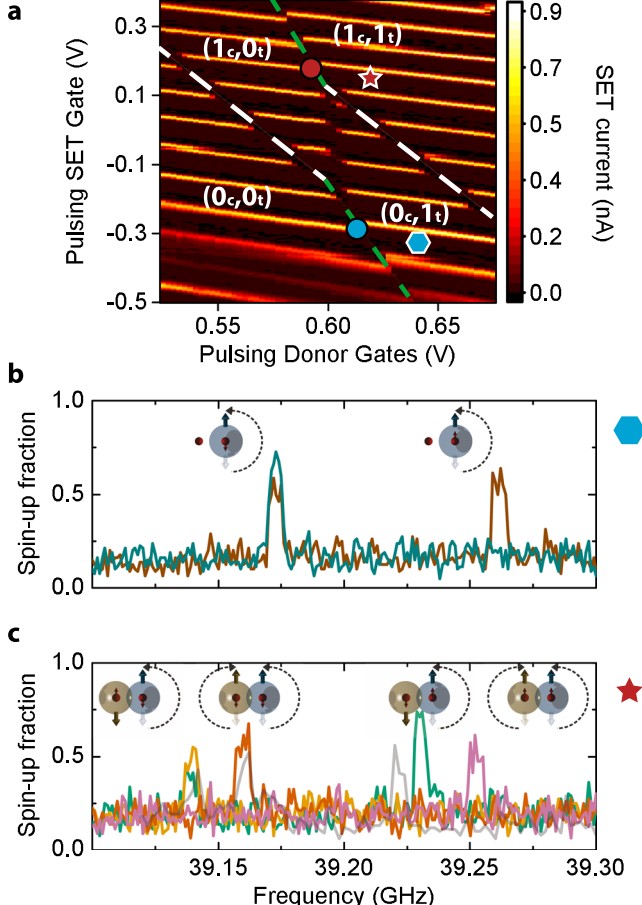

**Fig. 4 Signature of exchange coupling between electron spins in a $^{31}$P donor pair. a** Charge stability diagram around two donor charge transitions, obtained by scanning the voltages on the pulsing SET and the pulsing donor gates (unlike Fig. 3b, where the DC gates were scanned, which have stronger capacitive coupling to the donors). A clear two-electron honeycomb diagram can be resolved. The dashed white lines follow the control donor transition, while the dashed green lines follow the target donor. The measurement demonstrates an access to all charge occupation regions. Blue and red circles mark the spin readout points for the target electron, while the blue hexagon and red star mark the spin control regions for different charge occupations. **b** ESR spectrum acquired in the $(0_c,1_t)$ region (blue hexagon), i.e., with the control donor ionized. Only two ESR peaks arise, related to the nuclear spin configuration of the target donor. **c** If the ESR spectrum of the target donor is measured in the $(1_c,1_t)$ region (red star), the exchange coupling with the control electron gives rise to the four main peaks $\ell 1$ (yellow), $\ell 2$ (red), $\ell 4$ (green), and $\ell 5$ (pink), corresponding to the four possible nuclear spin configurations, while the control electron is $|\downarrow\rangle$. In one scan (gray line), we observed the occurrence of the rare $\ell 5_a$ transition (see Supplementary Note 1).

an exchange interaction $J$ between the two-donor electrons: the four ESR peaks correspond to the four possible orientations of the two donor nuclear spins, while the control donor is in the $|\downarrow\rangle$ state ($\ell 1$, $\ell 2$, $\ell 4$, and $\ell 5$). Observing all six main ESR lines would normally require preparing the control donor electron in the $|\uparrow\rangle$ state, which was not attempted in this experiment. Here, the nuclear spins' state was not deliberately controlled, but all spin configurations were eventually reached through random nuclear flips. In one occasion, we also detected an additional ESR peak, consistent with line $\ell 5_a$ (Fig. 4c, gray line). This resonance represents a (rare) transition from the two-electron $|T_-\rangle$ state to a state with a predominant $|S\rangle$ component, conditional on the $|\Uparrow_c\Uparrow_t\rangle$ nuclear spin configuration (see Supplementary Note 1).

Despite the 6 MHz width of the ESR lines caused by the adiabatic inversion, it is clear by comparing Fig. 4b, c that the addition of a second electron introduces a significant Stark shift of both the hyperfine coupling $A_t$ and the $g$-factor $g_t$ of the target donor. While Stark shifts of donor hyperfine couplings and $g$-factors as a function of applied electric fields have been observed before[38], including on single donors[39], the observation of such shifts from the addition of a single charge in close proximity is novel. We anticipate that a systematic analysis of $A$ and $g$ Stark shifts under controlled conditions may help elucidating the precise nature of the electron wavefunctions in exchange-coupled donors, and benchmarking the accuracy of microscopic models.

Once the approximate frequencies of the ESRs are found by adiabatic inversion with chirped pulses, we switch to short constant-frequency pulses in order to measure linewidths limited solely by the pulse excitation spectrum. Here, unlike the experiments in Fig. 4, the four different nuclear spin configurations $|\Downarrow_c\Downarrow_t\rangle, |\Downarrow_c\Uparrow_t\rangle, |\Uparrow_c\Downarrow_t\rangle, |\Uparrow_c\Uparrow_t\rangle$ are deliberately set by projective nuclear readout followed, if needed, by coherent manipulation of the individual nuclear spins with NMR pulses[35]. To address a specific nuclear spin, we keep the target donor ionized while the control donor is in the neutral state, with its electron spin in $|\downarrow\rangle$. This renders the NMR frequencies of each nucleus radically different, with $\nu_{nt} = \gamma_n B_0 \approx 24.173$ MHz and $\nu_{nc} = \gamma_n B_0 + A_c/2 \approx 67.92$ MHz (see Supplementary Note 5 for details on the nuclear spin initialization).

The full ESR spectrum is presented in Fig. 5b along with insets that display the individual power-broadened resonance peaks. The experimental ESR spectrum shown in Fig. 5b can be compared to the numerical simulations of the full Hamiltonian (Eq. (1)) for the specific parameters of this donor pair. In addition to the exchange coupling $J$, the Hamiltonian contains five unknown parameters: the contact hyperfine couplings $A_t$ and $A_c$, the electron $g$-factors $g_t$ and $g_c$, and the static magnetic field $B_0$. Although $B_0$ is imposed externally, its precise value at the donor sites can have a slight uncertainty, e.g., due to trapped flux in the superconducting solenoid, or positioning the device slightly off the nominal center of the field. $B_0$ can be combined with the average of $g_t$ and $g_c$ to yield an average of the Zeeman energy $\bar{E}_Z/h = (g_t + g_c)\mu_B B_0/2h$ of the donor electrons, which would rigidly shift the manifold of ESR frequencies. If, in addition, we assume that $g_t = g_c$, we are left with four free fitting parameters, $J, A_t, A_c, \bar{E}_Z/h$ which can be extracted from the knowledge of the four ESR frequencies.

In the numerical simulations, we vary the hyperfine coupling of target and control donors, $A_t$ and $A_c$, to find a combination of values that allows matching all four ESR frequencies at the same magnitude of the exchange interaction $J$. Figure 5a shows the result of the simulation that best matches the ESR spectrum of Fig. 5b, using $A_t = 97.75 \pm 0.07$ MHz, $A_c = 87.57 \pm 0.16$ MHz, and $J = 32.06 \pm 0.06$ MHz. Errors indicate the 95% confidence levels. With these values, all ESR frequencies were matched with a maximum error $\Delta\ell = \max(|\ell 1_{sim} - \ell 1_{exp}|; |\ell 2_{sim} - \ell 2_{exp}|; |\ell 4_{sim} - \ell 4_{exp}|; |\ell 5_{sim} - \ell 5_{exp}|) = 47.4$ kHz, only slightly larger than the 30 kHz resolution of the measurement itself. This spectroscopic method constitutes the most accurate measurement of exchange interaction between phosphorus donor pairs obtained to date.

The extracted values of $A$ are far from the bulk value $A_{bulk} = 117.53$ MHz and rather different between the two donors. This could be due to local variations in lattice strain and electric fields within the device, which can be substantial even on a scale $\approx 10$ nm. Strain, in particular, varies dramatically near the tips of the control gates[40], and is well-known to cause changes in hyperfine coupling[41,42]. The possible influence of strain on the spin relaxation time $T_1$ is discussed in ref. [43].

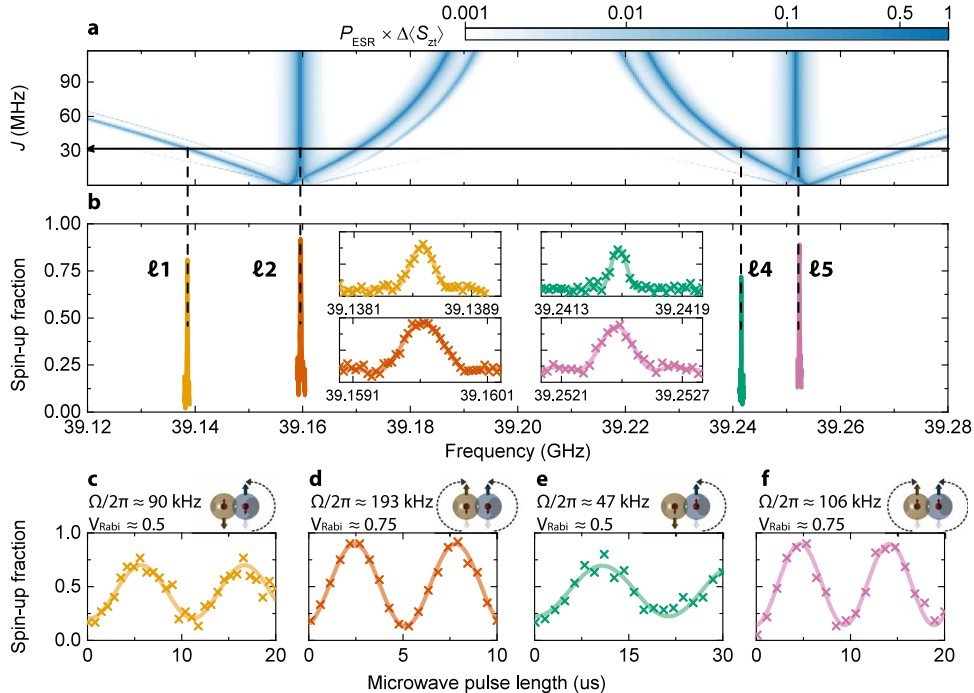

**Fig. 5 Conditional and unconditional coherent control of the target qubit in the presence of an exchange-coupled control qubit. a** Simulated evolution of the ESR spectrum as a function of exchange coupling $J$, using the system Hamiltonian (Eq. (1)) with parameters matching the experimental results. **b** Measured ESR spectrum of the target electron in the ($1_c1_t$) charge region. The control electron is kept in the $|\downarrow\rangle$ state, while the four nuclear spin configurations are deliberately initialized by nuclear magnetic resonance. All ESR peaks match the simulation by choosing the parameters $J = 32.06 \pm 0.06$ MHz, $A_t = 97.75 \pm 0.07$ MHz, $A_c = 87.57 \pm 0.16$ MHz, with maximum error $\Delta\ell = 47.4$ kHz. **c–f** Target qubit Rabi oscillations measured on each of the resonances, $\ell1$ (**c**), $\ell2$ (**d**), $\ell4$ (**e**), and $\ell5$ (**f**). A $\pi$-pulse on $\ell1$ or $\ell4$ transitions constitutes a CROT two-qubit logic gate (Fig. 2b). The same microwave source output power (8 dBm) has been used to drive all Rabi oscillations. The frequency $\Omega$ of the observed Rabi oscillations exhibits variations of up to a factor 4 between resonances, possibly due to a non-monotonic frequency response of the microwave transmission line. The visibility of the Rabi oscillations is systematically lower in the conditional resonances ($\ell1$ and $\ell4$), as compared to the unconditional ones ($\ell2$ and $\ell5$).

The observation of exchange coupling in the appropriate range for CROT operations was not unique to this particular device. A similar value of $J \approx 30$ MHz was measured on a second device, fabricated in the same batch and with the same high-dose P$^+$ implantation strategy (see Supplementary Note 6).

The high-fluence devices contain many donors, as visible in the charge stability diagram (Fig. 3b). One may thus expect to encounter more complex clusters of interacting donors, instead of isolated pairs. However, the high-resolution ESR spectrum in Fig. 5b, which does not contain extra resonances beyond those expected from a pure two-donor system, rules out other coupled donors. This is probably because the vast majority of other donors, presumably located behind (with respect to the SET) the pair being measured, are in the ionized charge state. To completely eliminate concerns around spurious donors, we will in the future adopt a counted single-ion implantation method, which allows to introduce individual donors with a confidence close to 99.9% (ref. [44]).

**Resonant CROT gate.** Coherent control of one of the two-electron spins is demonstrated in Fig. 5c–f. ESR control of the electron spin is performed in the ($1_c1_t$) region, with the control electron in the $|\downarrow\rangle$ state. We observe Rabi oscillations for all four nuclear spin configurations. Electron spin rotations driven on $\ell1$ ($|\Uparrow_c\downarrow_c\Downarrow_t\downarrow_t\rangle \leftrightarrow |\Uparrow_c\downarrow_c\Downarrow_t\uparrow_t\rangle$, yellow line) and $\ell4$ ($|\Downarrow_c\downarrow_c\Uparrow_t\downarrow_t\rangle \leftrightarrow |\Downarrow_c\downarrow_c\Uparrow_t\uparrow_t\rangle$, green line) are conditional upon the control electron being in the $|\downarrow\rangle$ state. Therefore, a $\pi$-pulse on one of these ESR resonances constitutes a CROT two-qubit gate.

For the "trivial" resonances, where the nuclear spins are either $|\Downarrow_c\Downarrow_t\rangle$ ($\ell2$, red line) or $|\Uparrow_c\Uparrow_t\rangle$ ($\ell5$, pink line), the Rabi oscillations have a visibility $V_{Rabi} = P_\uparrow(\pi) - P_\uparrow(0) \approx 0.75$. In contrast, the nontrivial, conditional resonances $\ell1$ and $\ell4$, have a significantly lower visibility $V_{Rabi} \approx 0.5$. We considered whether this could be explained by the fact that $\ell1$ and $\ell4$ represent transitions to the $\widetilde{|\downarrow\uparrow\rangle}$ state rather than $|\downarrow\uparrow\rangle$. Given the measured $J \approx 32.06$ MHz and $\bar{A} = 92.66$ MHz, the final state for resonances $\ell1$ and $\ell4$ is $\widetilde{|\downarrow_c\uparrow_t\rangle} = 0.986|\downarrow_c\uparrow_t\rangle + 0.166|\uparrow_c\downarrow_t\rangle$. This would account for only a 2.7% loss in visibility when measuring the transition through the target qubit.

Another possible contribution to the loss of Rabi visibility can arise because, in a coupled qubit system, measuring one qubit can affect the state of both. Here, the single-shot measurement of the target electron can result in the $\widetilde{|\downarrow_c\uparrow_t\rangle}$ state being projected to $|\downarrow_c\uparrow_t\rangle$ or $|\uparrow_c\downarrow_t\rangle$. If the system is projected to $|\uparrow_c\downarrow_t\rangle$ and the control electron is not reinitialized in $|\downarrow_c\rangle$ for the next single-shot measurement, the ESR resonances $\ell1$ or $\ell4$ become inactive. Resetting the control electron to the $|\downarrow\rangle$ state requires waiting a relaxation time $T_1$, during which no excitation of the target spin would be achieved on $\ell1$ or $\ell4$. In this device, we measured $T_1 = 3.4 \pm 1.3$ s on the target electron spin (Supplementary Note 3). Therefore, even though the chance of projection to $|\uparrow_c\downarrow_t\rangle$ is low (2.7%), this effect could propagate over several measurement records. This hypothesis can be verified by inspecting the single-shot readout traces (Supplementary Note 4). After a $\pi$-pulse on $\ell1$ or $\ell4$, we observe instances where a few successive readout traces show a $|\downarrow\rangle_t$ outcome. However, such instances of missing target excitation do not last for more than $\approx 20$ ms—two orders of magnitude less than the measured $T_1$ of the target

electron spin. Therefore, also this explanation appears improbable. Overall, we conclude that even performing a simple Rabi oscillation on a conditional resonance in exchange-coupled donors unveils unexpected details that warrant further investigation.

The presence within the device of strong electric fields, which can affect the value of $J$ and thereby the frequency of the conditional resonances, appears not to introduce spurious spin dephasing. We have measured the dephasing time of the target electron $T_2^*$ on both $\ell 5$ (unaffected by $J$) and $\ell 1$ (dependent on $J$), and found similar values $T_2^* \approx 9\,\mu s$, within the error margins (Supplementary Note 3).

The complete benchmarking of a two-qubit logic gate requires the coherent control and individual readout of both qubits, and the operation of the target qubit conditional on an arbitrary state of the control qubit. The present device, comprising a very thin ESR antenna[40], was damaged by an electrostatic discharge before we could complete the benchmarking of the full two-qubit logic gate. Future devices will be equipped with thicker antenna to prevent this issue.

For the readout, it is often but not always possible to read two (or more) spins sequentially using the same charge sensor. This depends simply on whether all donors electrons have a tunnel time to the reservoir that falls within a usable range (typically 10 µs–10 ms). We are currently developing new device designs, inspired by the flip-flop qubit proposal[45], that afford a greater degree of control of all tunnel couplings. Even if only one donor (e.g., the target) happens to be readable, the control donor spin states can be read out via a quantum non-demolition (QND) method by using the target electron as ancilla qubit, as already demonstrated in exchange-coupled double quantum dot systems[17,46,47]. This process requires a long relaxation time $T_1$ of the electron spins in presence of weak exchange coupling. The target electron $T_1 = 3.4 \pm 1.3$ s measured here is close to that of single, uncoupled donor electrons spins[43], and indicates that an ancilla-based QND readout will be an available option for future experiments.

## Discussion

We have presented the experimental observation of weak exchange coupling between the electron spins of a pair of $^{31}P$ donors implanted in $^{28}Si$. The exchange interaction $J = 32.06 \pm 0.06$ MHz was determined by ESR spectroscopy, and falls within the range $J < A$ where a native CROT two-qubit logic gate can be performed by applying a $\pi$-pulse to the target electron after setting the two donor nuclear spins in opposite states. These results represent the first demonstration of hyperfine-controlled CROT gate for donor electrons[13]—a scheme that is intrinsically robust to uncertainties in the donor location, since it only requires $J$ to be smaller than $A \approx 100$ MHz, and larger than the inhomgeneous ESR linewidth $\approx 10$ kHz.

The present work already unveiled peculiar effects, such as the Stark shift of hyperfine coupling and $g$-factors in the presence of an exchange-coupled electron, and unexplained features in the visibility of conditional qubit rotations. These effects call for detailed theoretical models of donor exchange under strain and electric fields, significantly expanding the existing theories of exchange in bulk silicon[23].

Future experiments will focus on benchmarking the fidelity of a complete one- and two-qubit gate set, and studying the noise channels affecting the operations. The suitability of this exchange-based logic gate for large-scale quantum computing will be assessed by integrating deterministic, counted single-ion implantation within the fabrication process[8,44], and studying the device yield and gate performance while subjected to realistic fabrication tolerances. The fine spectral resolution afforded by our resonant control methods will provide precious insights and experimental validation to a wide suite of theoretical models of donor physics and quantum device fabrication.

## Methods

**Sample fabrication**. Silicon MOS processes are employed for the donor spin qubit device fabrication. A silicon wafer is overgrown with a 0.9 µm thick epilayer of the isotopically purified $^{28}Si$ with $^{29}Si$ residual concentration of 730 p.p.m. (ref. [48]). Heavily doped n$^+$ regions for Ohmic contacts and lightly doped p regions for leakage prevention are defined by phosphorus and boron thermal diffusion. A field oxide (200 nm thick SiO$_2$) is grown using a wet thermal oxidation process. The central active area is covered with a high-quality thermal oxide (8 nm thick SiO$_2$) grown in dry conditions. Subsequently, an aperture of 90 nm × 100 nm is defined in a PMMA mask using electron-beam lithography (EBL). Through this aperture, the samples are implanted with atomic (P) or molecular (P$_2$) phosphorus ions at an acceleration voltage of 10 keV per ion. During implantation, the samples were tilted by 7° to minimize the possibility of channeling implantation. The final P atom position in the device is determined using full cascade Monte Carlo SRIM simulations. The projected range of the implant is ~10 nm beyond the SiO$_2$/Si interface. The size of the PMMA aperture is taken into account when determining the P–P donor spacing. Post implantation, a rapid thermal anneal (5 s at 1000 °C) is performed for donor activation and implantation damage repair. A nanoelectronic device is defined around the implantation region through two EBL steps, each followed by thermal deposition of aluminum (20 nm thickness for layer 1; 40 nm for layer 2). Between each aluminum layer, the Al$_2$O$_3$ is formed by immediate, post-deposition sample exposure to a pure, low pressure (100 mTorr) oxygen atmosphere. The final step is a forming gas anneal (400 °C, 15 min, 95% N$_2$/5% H$_2$) aimed at passivating the interface traps.

**Experimental setup**. The device was placed in a copper enclosure and wire-bonded to a gold-plated printed circuit board using thin aluminum wires. The sample was mounted in a Bluefors LD400 cryogen-free dilution refrigerator with base temperature of 14 mK, and placed in the center of the magnetic field produced by the superconducting solenoid in persistent mode (≈1.4 T). The magnetic field was oriented perpendicular to the short-circuit termination of the on-chip microwave antenna and parallel to the sample surface.

DC bias voltages, sourced from Stanford Instruments SIM928 isolated voltage sources, were delivered to the SET top gate, the barrier gates and the DC donor gates through 20 Hz low-pass filters. A room-temperature resistive combiner was used to add DC voltages (Stanford Instruments SIM928) to AC signals produced by a LeCroy ArbStudio 1104. The combined signals were delivered to the pulsing SET gate and the pulsing donor gates through 80 MHz low-pass filters. Microwave pulses for ESR were generated by an Agilent E8257D 50 GHz analog source; RF pulses for NMR were produced by a Agilent N5182B 6 GHz vector source. RF and microwave signals to be delivered to the microwave antenna were combined at room temperature and delivered through a semi-rigid coaxial cable fitted with a 10 dB attenuator mounted at the 4 K plate and a 3 dB attenuation at the 14 mK stage. The SET current was measured by a Femto DLPCA-200 transimpedance amplifier at room temperature (10$^7$ V/A gain, 50 kHz bandwidth), followed by a Stanford Instruments SIM910 JFET post-amplifier (10$^2$ V/V gain), Stanford Instruments SIM965 analog filter (50 kHz cutoff, low-pass Bessel filter), and acquired via an AlazarTech ATS9440 PCI digitizer card. The instruments were synchronized by a SpinCore Pulseblaster-ESR TTL generator.

## Data availability

The experimental and simulation data that support the findings of this study are available in Figshare with the identifier https://doi.org/10.6084/m9.figshare.13291913.

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

## Acknowledgements

The research at UNSW and U. Melbourne was funded by the Australian Research Council Centre of Excellence for Quantum Computation and Communication Technology (Grant No. CE170100012) and the US Army Research Office (Contract No. W911NF-17-1-0200). We acknowledge support from the Australian National Fabrication Facility (ANFF) and the AFAiiR node of the NCRIS Heavy Ion Capability for access to ion-implantation facilities. K. M.I. acknowledges support from the Spintronics Research Network of Japan. The views and conclusions contained in this document are those of the authors and should not be interpreted as representing the official policies, either expressed or implied, of the ARO or the US Government. The US Government is authorized to reproduce and distribute reprints for government purposes notwithstanding any copyright notation herein.

## Author contributions

M.T.M., and F.E.H. fabricated the devices, with A.M.'s and A.S.D.'s supervision. A.M.J., B.C.J., and D.N.J. designed and performed the ion implantation. K.M.I. supplied the isotopically enriched 28Si wafer. M.T.M. and A.L. performed the measurements and analyzed the data with A.M.'s supervision. M.T.M. and A.M. wrote the manuscript, with input from all coauthors.

## Competing interests

A.M. and A.L. are coauthors on the patent "Quantum Logic" (WO2014026222A1, AU2013302299B2, US20150206061A1, and EP2883194A4), which describes an implementation of two-qubit logic gates with donor spin qubits, some aspects of which are embodied in the results presented in this manuscript. Other authors declare no competing interests.
