## [Peer Review File · Nature Communications]

Reviewers' Comments:

Reviewer #1:

Remarks to the Author:

Summary of the work

The authors report the measurement of conditional Rabi oscillations between two exchange coupled electron spin qubits bound to two P donors in isotopically purified silicon. In short, this is the experimental demonstration of two qubit operation proposed earlier (ref 13) by the same group. They show that CROT operation is possible for exchange-coupled electron spins in the $J < A$ regime, and this scheme can be beneficial for alleviating positioning requirement in donor implantation since specific value of J is not important as long as $J < A$.

Assessment

The current experiment is indeed the first demonstration of CROT in the $J < A$ regime. Developments of simple and potentially high fidelity two qubit gate operation in semiconductor spin qubits is of high importance as the entire field is moving to developing multiple qubit operations nowadays. The proposed and demonstrated CROT operation is believed to be useful for building robust two qubit gate operations. I believe the method is also academically impactful as the implanted donor-based nuclear and electron spin qubits naturally forms two different types of spin qubits; fast and easy-to-read out electron spins and long-lived nuclear spins acting as quantum memory. The method requires that one should have concrete method to (1) place nearby exchange coupled spins (2) and deterministically prepare nuclear spins (parallel and/or anti-parallel) to detune electron spin splitting. The authors indeed achieve these requirements, and I believe the scheme is also compatible to other similar spin registers like defect color center-based spin registers so that the method can be insightful for broad range of spin-based quantum information platform research.

However, I believe the manuscript in the current form has some issues, which I list below.

Specific points

1) One of the main merits of the method is that the value of J has wide, a few orders of magnitude tolerance (lower bound: resonance line width, upper bound hyperfine strength). Can authors translate this to expected donor positioning tolerance and discuss whether it is well within current implantation capability? Perhaps the right place to add discussion is at the conclusion where the authors cite ref.8. I think this discussion is crucial to easily see the significance of the result.

2) Deterministically preparing nuclear spin states is crucial to realize CROT, but the information of experimental procedure is only discussed in a limited fashion. I understand that the related discussion is already preciously reported by the group, but I suggest to explain more on this, perhaps with appropriate pulse sequence figure.

3) In Fig. 5 c and e, only the oscillation Rabi oscillation is shown but it is important to also show that there is no oscillation when the control qubit is set to the other logical state, concretely showing that the operation is really conditional. From the precedent CROT mechanism analysis, it is believable that operations at I1, and I4 frequencies are conditional on the control qubit state, but experimentally showing that this is indeed the case is important. Similarly, operations at I3 and I6 frequencies are entirely missing. Is there any particular reason ?

4) Related to the point #3, the conditional operation is possible only for low enough Rabi driving amplitude (or Rabi frequency) as at high MW power even largely detuned states can be excited. With the proposed method, how fast one can realize, for example CNOT gate? I buy the main benefit of tolerable J value, but the authors should discuss expected two qubit gate speed that can potentially be disadvantage of this scheme.

5) In this type of devices, qubit-to-reservoir tunnel rates seems to be difficult to tune in- situ. I get

the point of potentially using quantum non-demolition measurement using achieved CROT as resource, but can this be extended to be used reading more than two spins with one sensor ?

5-1) Minor point related to #5: In ref 17., reading out second spin using CROT operation on spin 1 is actually performed, and I believe that the method is essentially quantum non-demolition measurement. I suggest to put ref 17, along with Refs 38, and 39.

To sum up, although I acknowledge the work's novelty (first demonstration), high degree of experimental achievement (exquisite multi spin coherent control), and wide applicability (general also to variety of electron-nuclear spin system), I hesitate to recommend the work for the publication in Nature Communications at least in the current form. The above points should be addressed before I reexamine my consideration.

Reviewer #2:

Remarks to the Author:

In the manuscript titled "Conditional quantum operation of two exchange-coupled single-donor spin qubits in a MOS-compatible silicon device", Mateusz Madzik et al. performed a meticulous study of a two-donor system in a SiMOS structure and demonstrated experimentally two-qubit dynamics conditional on the donor nuclear spin configurations. I find the study impressive in terms of the experimental capabilities, and the findings an important experimental milestone for donor-based spin qubits. While I am overall positive about this manuscript, there are a few issues that the authors should address before I can recommend its publication, as I discuss below.

I am not sure the title is accurate for this paper: to me "conditional quantum operation" hints at a degree of quantum control that the authors have not yet fully demonstrated. The authors have indeed been able to prepare the nuclear spins in desired orientations via NMR, and performed pulsed ESR on the target qubit. However, all the experiments are done with the control qubit in the down state. Wouldn't it be necessary to repeat these experiments under the condition that the control qubit be up to fully demonstrate differences in the target qubit dynamics in order to claim conditional operation? The present results are more conditional upon nuclear spin configurations rather than control qubit state. With the measured T1, it seems to me that keeping the control qubit in the up state and perform the same pulsed ESR experiments should not be dramatically more difficult than what the authors have already achieved. Without this part of the experiment, I think it would be more appropriate to use "conditional quantum dynamics" in the title of the paper.

The paper quoted hyperfine coupling strength of 97.75 and 87.57 MHz for the two donors. What are the reasons for such dramatic variation? The authors mentioned strain and electric field, but are there references to show that A can be this different due to strain? To put it another way, can two donors separated by maybe 20 nm experience hydrostatic strain that is different by this much? If electric field contribute to this variation, wouldn't it also lead to possibly charge noise induced dephasing for the spin qubits? Are the measured dephasing times consistent with bulk values, or significantly shortened to support this scenario? Some more careful discussions would certainly help give readers more food for thought.

The current device is fabricated via ion implantation with high fluence. According to Fig.2 of the Supplementary Information, it seems that the interdonor distance would be on average below 10 nm. With such a small donor separation I would expect the exchange coupling to be quite large instead of only 32 MHz, or 8 neV. Maybe this particular pair of donors do have a much larger separation, or their coupling is significantly suppressed by valley interference. Either way, with the high fluence of ions coming in, shouldn't one expect the presence of other donors that have significant exchange coupling with either or both donors? Maybe those donors cannot be detected directly by the approach adopted in this study, but surely they would influence the dynamics

and/or coherence of the two donors? Are there ways this issue can be clarified?

The authors mentioned several interesting issues in the conclusion. I believe a more detailed discussion would better lay out the open problems and possibly attract the attention of more researchers to study the donor spin qubits.

In summary, I find the reported experimental study really impressive, and the paper well written. If the issues raised above can be properly addressed the paper should be ready for publication.

Reviewer #3:

Remarks to the Author:

This manuscript by Madzik et al. shows a series of experiments with donor-bound, exchange-coupled electron spins in a Si-MOS double quantum dot. The authors presented new ion implantation strategies and measured the exchange coupling in their devices to be on the order of 100 MHz. This value allows the authors to drive Rabi rotations of one electron conditional on the state of the other, a first step toward realizing a CROT two-qubit gate in their device. The characterization techniques are very standard in this manuscript and the results are convincing. However, I do have one comment which I hope the authors could address before a decision on publication is made

Two-qubit gates for donor-based spin qubits have always been a bottleneck for the community owing to the fact that the exchange coupling can fluctuate by orders of magnitude even when the atom is moved by just one lattice site (a point the authors also made in the introduction). In my opinion, a true step forward in addressing this challenge is to show a fabrication process that can robustly produce exchange coupling on the order of 100 MHz. To demonstrate this, it is necessary to at least present data from more than one device. I'm sure the authors must have measured other devices from this new fabrication process. Could they summarize what the exchange couplings are in the other devices? My concern is that this particular device in the manuscript is just one outlier device that "happened to work". If the authors can alleviate this concern by showing data from other devices, I have no problem with recommending publication.

We thank the Referees for their careful reading of our manuscript, and for providing many recommendations to improve its clarity and impact.

Below we provide a point-by-point response to their comments and queries. **Our replies are marked in blue. Descriptions of changes to the manuscript are marked in red, and also highlighted in red within the revised manuscript.**

In the replies below, cited references are the ones in the updated manuscript (the numbering is potentially different from the one in the first submission, due to new references added).

Reviewer #1:

Summary of the work

The authors report the measurement of conditional Rabi oscillations between two exchange coupled electron spin qubits bound to two P donors in isotopically purified silicon. In short, this is the experimental demonstration of two qubit operation proposed earlier (ref 13) by the same group. They show that CROT operation is possible for exchange-coupled electron spins in the $J < A$ regime, and this scheme can be beneficial for alleviating positioning requirement in donor implantation since specific value of J is not important as long as $J < A$.

We thank the Referee for this clear and accurate description of the essence of our work.

Assessment

The current experiment is indeed the first demonstration of CROT in the $J < A$ regime. Developments of simple and potentially high fidelity two qubit gate operation in semiconductor spin qubits is of high importance as the entire field is moving to developing multiple qubit operations nowadays. The proposed and demonstrated CROT operation is believed to be useful for building robust two qubit gate operations. I believe the method is also academically impactful as the implanted donor-based nuclear and electron spin qubits naturally forms two different types of spin qubits; fast and easy-to-read out electron spins and long-lived nuclear spins acting as quantum memory. The method requires that one should have concrete method to (1) place nearby exchange coupled spins (2) and deterministically prepare nuclear spins (parallel and/or anti-parallel) to detune electron spin splitting. The authors indeed achieve these requirements, and I believe the scheme is also compatible to other similar spin registers like defect color center-based spin registers so that the method can be insightful for broad range of spin-based quantum information platform research.

However, I believe the manuscript in the current form has some issues, which I list below.

Specific points

1) One of the main merits of the method is that the value of J has wide, a few orders of magnitude tolerance (lower bound: resonance line width, upper bound hyperfine strength). Can authors translate this to expected donor positioning tolerance and discuss whether it is well within current implantation capability? Perhaps the right place to add discussion is at the conclusion where the authors cite ref.8. I think this discussion is crucial to easily see the significance of the result.

We completely agree that such discussion would greatly strengthen the significance of our result. However, answering the Referee's question is chiefly a theory/modelling effort - the validity of the

answer is only as good as the accuracy of the model used to arrive at it. Because we are working with nanoscale electronic devices where the donors are subjected to large electric fields, it is not sufficient to recycle the calculations of exchange vs. distance performed over the last 20 years for donor pairs in bulk silicon. Therefore, we are currently collaborating with colleagues at Sandia National Labs to develop a full-configuration interaction, effective-mass code that will allow us to efficiently compute J as a function of distance (in all directions) *and* as a function of electric field. This effort is progressing well but its results are not ready for being discussed here. They will be reported in a separate theoretical paper in the very near future.

We now briefly mention future directions for device modelling in the conclusions, page 8.

2) Deterministically preparing nuclear spin states is crucial to realize CROT, but the information of experimental procedure is only discussed in a limited fashion. I understand that the related discussion is already preciously reported by the group, but I suggest to explain more on this, perhaps with appropriate pulse sequence figure.

We thank the Referee for this suggestion. We now extensively explain the nuclear initialization in a new section in the Supplementary Information – Section V. The new Supplementary Figure 5 provides a detailed description of the pulse sequence.

3) In Fig. 5 c and e, only the oscillation Rabi oscillation is shown but it is important to also show that there is no oscillation when the control qubit is set to the other logical state, concretely showing that the operation is really conditional. From the precedent CROT mechanism analysis, it is believable that operations at 11, and 14 frequencies are conditional on the control qubit state, but experimentally showing that this is indeed the case is important. Similarly, operations at 13 and 16 frequencies are entirely missing. Is there any particular reason ?

Yes, there is a particular (unfortunate) reason. This batch of devices was fabricated with an on-chip microwave antenna which we chose to equip with a very thin (~50x50 nm cross-section) short-circuit termination, in order to maximize the oscillating magnetic field produced at the donor site. As a result, these devices turned out to be very fragile. The device used for the present work ended up being damaged by a lightning strike before we could perform the experiments that the Referee suggests. A current spike through the antenna termination melted it (like a fuse) and turned it into an open circuit. Such damaged antenna can no longer deliver oscillating magnetic fields at the tens of MHz frequencies necessary to control the nuclear spins. It is still able to create some oscillating magnetic field at 40 GHz for ESR, albeit with reduced amplitude, but the inability to control the nuclei makes the experiment essentially unfeasible.

As a side note, a similar accident also led to the discovery of Nuclear Electric Resonance in a single ^{123}Sb nucleus [S. Asaad et al., Nature 579, 205 (2020)]. In that paper, we presented an extensive discussion of the impact of such accidents, including a scanning electron micrograph of a damaged device. We now cite [S. Asaad et al., Nature 579, 205 (2020), Ref. 40] and briefly discuss the impact of electrostatic discharge in these devices, on page 8, and in Supplementary Section VI.

4) Related to the point #3, the conditional operation is possible only for low enough Rabi driving amplitude (or Rabi frequency) as at high MW power even largely detuned states can be excited. With the proposed method, how fast one can realize, for example CNOT gate? I buy the main benefit of tolerable J

value, but the authors should discuss expected two qubit gate speed that can potentially be disadvantage of this scheme.

We thank the Referee for suggesting to discuss this point. It was treated extensively in Ref. [Kalra14], but it is indeed useful to remind the reader of the key points. At the simplest level, the CNOT gate speed is limited by J , in the sense that one cannot use pulses so short that their spectral width encompasses other resonances (doing so would render the rotation no longer conditional on the state of the control qubit). There do exist clever pulse designs, which allow to drive the CNOT gate faster than the trivial limit would suggest. These would become important if one were to operate in a device where J is small. Here, with a $J > 30$ MHz, the issue did not arise (the fastest Rabi frequencies we have ever achieved with these devices were around 3 MHz – see Ref. [Pla12]).

We have added a brief discussion of this point on page 3, and two references [28,29] that discuss strategies to circumvent the spectral broadening limit.

5) In this type of devices, qubit-to-reservoir tunnel rates seems to be difficult to tune in-situ. I get the point of potentially using quantum non-demolition measurement using achieved CROT as resource, but can this be extended to be used reading more than two spins with one sensor ?

Using more than two spins with one sensor would require operating 3-qubit Toffoli gates instead of CROT gates. This might be possible, but we do not mean to imply that it is the best or the only way forward. We mentioned the use of CROT for QND measurements simply as a useful side-effect of what we demonstrated here. In parallel, we are attacking the problem of reading out multiple donors from other directions, namely by designing new device layouts that incorporate a tunnel-rate gate between donors and SET. We have seen early evidence of about an order of magnitude tunability in the tunnel rate. This is being done under the umbrella of flip-flop qubit device development (the proposal described in the new Ref. [44]) and will be published alongside the experimental demonstration of that kind of device.

We now briefly discuss strategies for tuning the qubit-to-reservoir tunnel rate on page 8.

5-1) Minor point related to #5: In ref 17., reading out second spin using CROT operation on spin 1 is actually performed, and I believe that the method is essentially quantum non-demolition measurement. I suggest to put ref 17, along with Refs 38, and 39.

The Referee is correct, Ref. 17 indeed measured Qubit 1 via Qubit 2 using a CROT gate, precisely as we suggest. We have added that reference alongside 38 and 39.

To sum up, although I acknowledge the work's novelty (first demonstration), high degree of experimental achievement (exquisite multi spin coherent control), and wide applicability (general also to variety of electron-nuclear spin system), I hesitate to recommend the work for the publication in Nature Communications at least in the current form. The above points should be addressed before I reexamine my consideration.

We thank the Referee for their positive appraisal of the novelty, quality and wide impact of our work; we hope to have addressed their points in their totality.

Reviewer #2

In the manuscript titled "Conditional quantum operation of two exchange-coupled single-donor spin qubits in a MOS-compatible silicon device", Mateusz Madzik et al. performed a meticulous study of a two-donor system in a SiMOS structure and demonstrated experimentally two-qubit dynamics conditional on the donor nuclear spin configurations. I find the study impressive in terms of the experimental capabilities, and the findings an important experimental milestone for donor-based spin qubits.

We thank the Referee for their enthusiastic appraisal of the quality of our work.

While I am overall positive about this manuscript, there are a few issues that the authors should address before I can recommend its publication, as I discuss below.

I am not sure the title is accurate for this paper: to me "conditional quantum operation" hints at a degree of quantum control that the authors have not yet fully demonstrated. The authors have indeed been able to prepare the nuclear spins in desired orientations via NMR, and performed pulsed ESR on the target qubit. However, all the experiments are done with the control qubit in the down state. Wouldn't it be necessary to repeat these experiments under the condition that the control qubit be up to fully demonstrate differences in the target qubit dynamics in order to claim conditional operation? The present results are more conditional upon nuclear spin configurations rather than control qubit state. With the measured T_1 , it seems to me that keeping the control qubit in the up state and perform the same pulsed ESR experiments should not be dramatically more difficult than what the authors have already achieved. Without this part of the experiment, I think it would be more appropriate to use "conditional quantum dynamics" in the title of the paper.

This comment is entirely accurate and aligns with point 3 of Reviewer #1. As explained in the response to that comment, a lightning strike damaged the device before we were able to actually perform a rotation of the target qubit conditional on both states of the control. We now explain this matter on page 8, and Supplementary Section VI.

We hope the Referees appreciated our careful choice of words. We did not title the paper "Demonstration of two-qubit CROT gate" or something to that effect. We called it "Conditional quantum operation" instead, because we indeed operated the target qubit conditional on a specific state of the control. Had we operated it conditional on an arbitrary state, we would claim to have demonstrated a universal two-qubit logic. We feel that the term "quantum dynamics" would not improve the accuracy of the description: "dynamics" usually means free evolution under some static Hamiltonian. Here, we applied deliberate coherent microwave control pulses, which we think are best described by the term "control".

The paper quoted hyperfine coupling strength of 97.75 and 87.57 MHz for the two donors. What are the reasons for such dramatic variation? The authors mentioned strain and electric field, but are there references to show that A can be this different due to strain?

The effect of strain on the hyperfine coupling is well documented in experiments on bulk donors. We have added a discussion of strain in page 6 of the main manuscript, and included two references that describe the effect of strain in silicon devices.

To put it another way, can two donors separated by maybe 20 nm experience hydrostatic strain that is different by this much?

We thank the Referee for raising this point. The lattice strain (not necessarily hydrostatic) does change dramatically over a few nanometers in our devices, because it is caused by differential thermal contraction of our 30-nm wide aluminium gates. We provided an extensive discussion – along with detailed finite-element modelling – of device strain in [S. Asaad et al., Nature 579, 205 (2020)]. We now cite that paper and include, on page 6, a brief discussion of the role of strain effects in our experiment.

If electric field contribute to this variation, wouldn't it also lead to possibly charge noise induced dephasing for the spin qubits? Are the measured dephasing times consistent with bulk values, or significantly shortened to support this scenario? Some more careful discussions would certainly help give readers more food for thought.

Electric fields of the order of many MV/m are indeed present in our devices. Their effect on qubit coherence is still somewhat unclear. We obviously take every precaution to ensure that the voltages on the gates are carefully filtered from technical noise. More doubtful is whether electric field fluctuations caused by interface charge traps have an effect on dephasing time. Early experiments [J. Muhonen et al., Nature Nano. 9, 986 (2014), Ref. 12] indicated that the dephasing time is not limited by charge noise. The present experiment seems to further reinforce this conclusion, because we see no difference in dephasing times between the unconditional resonance I5 and the conditional resonance I1. This was already discussed in Supplementary Section III. We have now briefly reiterated this discussion on page 7 of the main manuscript.

The current device is fabricated via ion implantation with high fluence. According to Fig.2 of the Supplementary Information, it seems that the interdonor distance would be on average below 10 nm. With such a small donor separation I would expect the exchange coupling to be quite large instead of only 32 MHz, or 8 neV. Maybe this particular pair of donors do have a much larger separation, or their coupling is significantly suppressed by valley interference. Either way, with the high fluence of ions coming in, shouldn't one expect the presence of other donors that have significant exchange coupling with either or both donors? Maybe those donors cannot be detected directly by the approach adopted in this study, but surely they would influence the dynamics and/or coherence of the two donors? Are there ways this issue can be clarified?

The Referee is quite correct in expecting that, with the fluence we have used, there should be several other donors in close proximity to the ones measured in this experiment. This issue can be addressed in a number of ways:

- When we tune the electrostatic landscape of the device to “zoom in” the desired donors, we use voltages such that one particular donor (the target, in this case) is close to the Fermi level in the device. The vast majority of other donors, which are farther back with respect to the SET reservoir, are in ionized state under these conditions. Therefore, even though they might “exist”, their presence is inconsequential. The hyperfine coupling of between the electron on a donor and the nucleus of another donor ~10 nm away is minuscule and we have never observed any evidence for it.

- We can further verify that no other donors need to be taken into account by inspecting the ESR spectrum. Other exchange-coupled donors would introduce more resonances in the spectrum. Thanks to the excellent dephasing times, we have a < 10 kHz spectral resolution, which allows us to conclude that no other donors are coupled to the ones under measurement – at least to within less than 10 kHz strength.

- For the future, we are gearing up to fabricate and operate devices using counted single-ion implantation. We have recently demonstrated the ability to count a single ³¹P atom with 99.87% confidence [A. Jakob

et al., arXiv:2009.02892]. With the ability to count precisely how many donors we implant, we will rule out with certainty the issue raised here.

We have added a discussion of the effects of other donors, and the perspective of removing such issues using counted ions, on page 6.

The authors mentioned several interesting issues in the conclusion. I believe a more detailed discussion would better lay out the open problems and possibly attract the attention of more researchers to study the donor spin qubits.

We have expanded the topics mentioned in the conclusions, particularly around future theoretical models, and the impact that our work can have on their development.

In summary, I find the reported experimental study really impressive, and the paper well written. If the issues raised above can be properly addressed the paper should be ready for publication.

We thank the Referee for the positive appraisal of our work, and we hope to have thoroughly addressed the issues raised.

Reviewer #3

This manuscript by Madzik et al. shows a series of experiments with donor-bound, exchange-coupled electron spins in a Si-MOS double quantum dot. The authors presented new ion implantation strategies and measured the exchange coupling in their devices to be on the order of 100 MHz. This value allows the authors to drive Rabi rotations of one electron conditional on the state of the other, a first step toward realizing a CROT two-qubit gate in their device. The characterization techniques are very standard in this manuscript and the results are convincing. However, I do have one comment which I hope the authors could address before a decision on publication is made

Two-qubit gates for donor-based spin qubits have always been a bottleneck for the community owing to the fact that the exchange coupling can fluctuate by orders of magnitude even when the atom is moved by just one lattice site (a point the authors also made in the introduction). In my opinion, a true step forward in addressing this challenge is to show a fabrication process that can robustly produce exchange coupling on the order of 100 MHz. To demonstrate this, it is necessary to at least present data from more than one device. I'm sure the authors must have measured other devices from this new fabrication process. Could they summarize what the exchange couplings are in the other devices? My concern is that this particular device in the manuscript is just one outlier device that "happened to work". If the authors can alleviate this concern by showing data from other devices, I have no problem with recommending publication.

The Referee is touching upon a key issue, not just for us, but for the semiconductor qubit community at large. In general, measuring and characterizing semiconductor devices is a very time-consuming exercise, which requires blocking a dilution fridge for several months. This is why it is exceedingly rare to find data from more than one device presented in a paper – not just with donors, but also with quantum dots.

However, “fortunately” (or rather, unfortunately but with a positive side-effect for this discussion) we do have data from a second device. As already explained in the response to Referees #1 and #2, this batch of devices was fabricated with a very thin ESR antenna termination, which makes the devices prone to

electrical damage. Before measuring the device on which all the reported experiments were conducted, we tested another device, which appeared to have a damaged ESR antenna from the very start. With that, we could still drive ESR transitions at 40 GHz, but we could not control the nuclei at tens of MHz. We took a number of ESR spectra with adiabatic inversion and were lucky enough to see occurrences of what we interpret as 11 and 13, separated by $J \sim 30$ MHz, thus very similar to the subsequent device. Because we could not control the nuclei, we abandoned further experiments.

We have added a new Supplementary Section VI which briefly discusses the data from this early device, and included the ESR spectrum that shows J-coupling of order 30 MHz. This is by no means a significant statistical sample, but we hope it provides the Referee with some reassurance that what we showed is not just a device that “happened to work”.

Reviewers' Comments:

Reviewer #1:

Remarks to the Author:

The authors have successfully addressed all of my comments and suggestions. I am satisfied with the revisions such as new references, detailed experimental sequence, and discussion of future strategies. I believe they also have served to address the other Referees' comments properly.

I now understand the reason behind the lack of experimental data performed at all the resonance frequencies. As a researcher in the same field, I understand the difficulty of this kind of experiment to get complete set of data before the device ends up not working. Even with this limitation, I evaluate that the data the authors gathered so far demonstrates successfully conditional nature of the quantum operation in $J < A$ regime, and the authors paid careful attention to choose right wording for the title : "conditional operation" rather than "full two qubit operation". Also with data on the second device, I believe that the main point of the work is convincing.

Thus, I now recommend that paper for the publication. I have no more comments.

Reviewer #2:

Remarks to the Author:

I am satisfied with the authors' replies to my questions and comments, and the changes they have implemented. As such I recommend the publication of this paper in Nature Communications.

I have one comment that I think the authors should consider to address in any further revision. As they pointed out the device they study has pretty strong variability in local strain, as evidenced by the large difference in hyperfine coupling strength. Have they considered other donor electron properties that may have also been impacted? For example g-factor? Or spin relaxation rate? Comments on such properties, while not impacting the current work, could provide insight into the long term viability of donor-based qubits.

Reviewer #3:

Remarks to the Author:

The authors have addressed my concerns satisfactorily and I recommend the manuscript for publication.

Below we provide a point-by-point response to their comments and queries. **Our replies are marked in blue. Descriptions of changes to the manuscript are marked in red, and also highlighted in red within the revised manuscript.**

Reviewer #1 (Remarks to the Author):

The authors have successfully addressed all of my comments and suggestions. I am satisfied with the revisions such as new references, detailed experimental sequence, and discussion of future strategies. I believe they also have served to address the other Referees' comments properly.

I now understand the reason behind the lack of experimental data performed at all the resonance frequencies. As a researcher in the same field, I understand the difficulty of this kind of experiment to get complete set of data before the device ends up not working. Even with this limitation, I evaluate that the data the authors gathered so far demonstrates successfully conditional nature of the quantum operation in $J < A$ regime, and the authors paid careful attention to choose right wording for the title : "conditional operation" rather than "full two qubit operation". Also with data on the second device, I believe that the main point of the work is convincing.

Thus, I now recommend that paper for the publication. I have no more comments.

We thank the Referee for their kind words, especially realizing our care in wording the nature of our achievement.

Reviewer #2 (Remarks to the Author):

I am satisfied with the authors' replies to my questions and comments, and the changes they have implemented. As such I recommend the publication of this paper in Nature Communications.

I have one comment that I think the authors should consider to address in any further revision. As they pointed out the device they study has pretty strong variability in local strain, as evidenced by the large difference in hyperfine coupling strength. Have they considered other donor electron properties that may have also been impacted? For example g-factor? Or spin relaxation rate? Comments on such properties, while not impacting the current work, could provide insight into the long term viability of donor-based qubits.

The impact of strain on other donor's properties has been rarely discussed in the literature, except for a brief discussion in a paper from our own group, Ref. [43], which analysed donor spin relaxation rates in several devices.

We have added a mention of Ref. [43] in relation to the effect of strain on donor spin relaxation.

Reviewer #3 (Remarks to the Author):

The authors have addressed my concerns satisfactorily and I recommend the manuscript for publication.

We thank the Referee for this recommendation.